# Selection of reference genes for quantitative RT-PCR (RT-qPCR) analysis of rat tissues under physiological and toxicological conditions

Terje Svingen, Heidi Letting, Niels Hadrup, Ulla Hass and Anne Marie Vinggaard

Division of Toxicology and Risk Assessment, National Food Institute, Technical University of Denmark, Søborg, Denmark

## ABSTRACT

In biological research the analysis of gene expression levels in cells and tissues can be a powerful tool to gain insights into biological processes. For this, quantitative RT-PCR (RT-qPCR) is a popular method that often involve the use of constitutively expressed endogenous reference (or 'housekeeping') gene for normalization of data. Thus, it is essential to use reference genes that have been verified to be stably expressed within the specific experimental setting. Here, we have analysed the expression stability of 12 commonly used reference genes (Actb, B2m, Gapdh, Hprt, Pgk1, Rn18s, Rpl13a, Rps18, Rps29, Sdha, Tbp and Ubc) across several juvenile and adult rat tissues (liver, adrenal, prostate, fat pad, testis and ovaries), both under normal conditions and following exposure to various chemicals during development. Employing NormFinder and BestKeeper softwares, we found Hprt and Sdha to be amongst the most stable genes across normal and manipulated tissues, with several others also being suitable for most tissues. Tbp and B2m displayed highest variability in transcript levels between tissues and developmental stages. It was also observed that the reference genes were most unstable in liver and testis following toxicological exposure. For future studies, we propose the use of more than one verified reference gene and the continuous monitoring of their suitability under various experimental conditions, including toxicological studies, based on changes in threshold (Ct) values from cDNA samples having been reverse-transcribed from a constant input concentration of RNA.

Corresponding author
Terje Svingen, tesv@food.dtu.dk

## INTRODUCTION

Despite the advent of high-throughput methods such as RNA-sequencing to measure transcript abundance in cells and tissues, quantitative RT-PCR (RT-qPCR) remains the method of choice for many, particularly when only a selected number of genes are to be analysed. The RT-qPCR methodology has itself improved greatly over the past two decades with advances to most parameters, including reaction chemistries and platform

technologies. However, it remains critically dependent upon a number of technical parameters that can significantly impact on the end results (*Nolan, Hands & Bustin, 2006*). Among the most critical aspects is correct normalisation, both to control for experimental errors and adjust for inter-sample variations.

There are many strategies for normalising RT-qPCR data (*Huggett et al., 2005*) and the most frequently used is normalisation with an endogenous reference gene that is stably expressed across all the samples. This approach using the comparative Ct (or $2^{-\Delta Ct}$) method (*Schmittgen & Livak, 2008*) is sound, but rests on the assumption that transcript abundance of the reference gene is the same in all cells, under all conditions. A growing body of evidence has clearly shown this not to be the case and that 'stably expressed' reference genes must be determined for each cell/tissue type during different stages of development and under all experimental conditions (reviewed by *Huggett et al., 2005*; *Kozera & Rapacz, 2013*). This can quickly become a Herculean task and likely a major reason why it remains so frequently overlooked. In a recent report, Piller and co-workers scrutinized recent publications in their field of research—rat spared nerve injury model of neuropathic pain—for evidence-based use of reference genes in RT-qPCR experiments and found that only 2 out of 26 peer-reviewed articles referred to proper validation for their reference genes (*Piller, Decosterd & Sutler, 2013*).

Over the years, there has been a steady increase in studies reporting on the suitability of various reference genes in rat tissues under specific experimental conditions. The liver has perhaps received most attention, but also include tissues such as the intestine, brain and other neuronal tissues, muscle, spleen, mammary gland, lung, and ovary (*Cabiati et al., 2012*; *Das, Banerjee & Shapiro, 2013*; *DuBois et al., 2013*; *Hvid et al., 2011*; *Langnaese et al., 2008*; *Lardizábal et al., 2012*; *Martínez-Beamonte et al., 2011*; *Pohjanvirta et al., 2006*; *Taki, Abdel-Rahman & Zhang, 2014*; *Verma & Shapiro, 2006*). What is striking about these studies, and others, is the large overlap in genes that have been analysed, but relatively small overlap in what genes are deemed expressed stable enough to be used for normalization purposes. The most commonly used genes are *Actb*, *Gapdh*, *Rn18s* (18S rRNA), *Hprt1* and *B2m*, perhaps because they are historical carry-overs from other semi-quantitative methods and not because they have been empirically tested. In fact, these genes have often been shown not to be particularly stable under various experimental conditions, but nevertheless continue to be the most frequently used.

In animal-based toxicological studies, the rat has become a tractable model and gene expression profiling is often performed. Studies also often include an array of different tissues, perhaps from various stages of development. Thus, these tissues vary not only in cell composition, but also by experimental manipulations, and the variability of tissues may potentially extend to expression levels of endogenous reference genes. It is now clear that commonly used reference genes such as *Actb*, *Gapdh*, *Ubc* and *Rn18s* can vary considerably depending on tissue types, developmental stage, sex, pathology, and experimental conditions (*Das, Banerjee & Shapiro, 2013*; *Kim et al., 2011*; *Martínez-Beamonte et al., 2011*; *Pohjanvirta et al., 2006*; *Ruedrich et al., 2013*; *Swijsen et al., 2012*). Thus, more emphasis should be given to proper validation of suitable reference genes to ensure accurate,

reproducible and biologically relevant gene expression data. Here, we have addressed this issue by analysing the expression stability of 12 putative endogenous reference genes in rat tissues, both from unexposed controls and from rats having been exposed to chemicals during development.

# MATERIALS AND METHODS

## Animals

Experimental protocols and use of animals were approved by the Danish Animal Experiments Inspectorate (Permit No. 2012-15-2934-00089 C4) and overseen by the Animal Welfare Committee of the National Food Institute, Technical University of Denmark. All tissue samples used in this study were from Wistar rats, either control rats, or rats having been exposed to a mixture of chemicals as described previously (*Christiansen et al., 2012*; *Hadrup et al., 2015*). In short, one group was exposed perinatally to a mixture of 13 known endocrine disrupting compounds at a dose estimated at 450-times higher than that of human exposure, designated Mix450 (*Christiansen et al., 2012*), with juvenile tissue samples collected on postnatal days (P): livers on P13; testis, prostate and adrenal on P16; ovaries on P17; and adult tissues after P55. A second and third group of juvenile male rats was exposed to 5 mg/kg perfluoronanoic acid (PFNA) and 5 mg/kg PFNA in addition to a mixture of 14 chemicals (PFNA/mix), respectively (*Hadrup et al., 2015*), and tissues were collected from adult rats.

## RNA extraction, cDNA synthesis and quantitative RT-PCR (RT-qPCR)

Total RNA was extracted from homogenized rat tissues using the RNeasy Mini kit (Qiagen, HIlden, Germany) including on-column DNaseI treatment. RNA purity and quantity were measured by nano-drop spectrophotometry, and 500 ng total RNA (A260/280 ratio of $1.95 \pm 0.1$) used to synthesise cDNA in the presence of 6 µM Random Primer mix (New England Biolabs, Ipswitch, Massachusetts USA) using the Omniscript kit (Qiagen, HIlden, Germany) in 20 µl reactions as per manufacturer's instructions. cDNA samples were diluted 1:20 and 3 µl used in 11 µl RT-qPCR reactions together with 5 µl TaqMan Fast Universal Master mix (Life Technologies, Carlsbad, California, USA), 0.5 µl TaqMan Gene Expression Assay (Life Technologies, Carlsbad, California, USA) and 2.5 µl sterile water. RT-qPCR assays were run in duplicates on a 7900HT Fast Real-Time PCR System (Applied Biosystems, Carlsbad, California, USA) in 384-well plates over 45 cycles of 95 °C for 1 s and 60 °C for 20 s in a two-step thermal cycle preceded by an initiation step of 95 °C for 20 s. Accompanying software was used for the acquisition of threshold cycle (Ct) values. Individual TaqMan Gene Assays with verified amplification efficiencies were purchased from Life Technologies and their corresponding product numbers are listed in Table 1. The *Rn18s* assay was designed previously (*Laier et al., 2006*), with forward and reverse primers run at 900 nM and TaqMan probe at 250 nM final concentrations. Amplification efficiency of the *Rn18s* assay was calculated to 97% by standard curve analysis on 6 serial 10-fold dilutions in triplicates.

**Table 1 List of putative rat reference genes and corresponding TaqMan assays.**

| Gene | RefSeq | Name | TaqMan assay |
|---|---|---|---|
| *Actb* | NM_031144 | Beta actin | Rn00667869_m1 |
| *B2m* | NM_012512 | Beta-2 microglobulin | Rn00560865_m1 |
| *Gapdh* | NM_017008 | Glyceraldehyde-3-phosphate dehydrogenase | Rn01775763_g1 |
| *Hprt1* | NM_012583 | Hypoxanthine guanine phosphoribosyl transferase | Rn01527840_m1 |
| *Pgk1* | NM_053291 | Phosphoglycerate kinase 1 | Rn01474008_gH |
| *Rn18s* | NR_046237 | 18S ribosomal RNA | (*Laier et al., 2006*) |
| *Rpl13a* | NM_173340 | Ribosomal protein L13A | Rn00821946_g1 |
| *Rps18* | NM_213557 | Ribosomal protein S18 | Rn01428913_gH |
| *Rps29* | NM_012876 | Ribosomal protein S29 | Rn00820645_g1 |
| *Sdha* | NM_130428 | Succinate dehydrogenase complex, subunit A, flavoprotein (Fp) | Rn00590475_m1 |
| *Tbp* | NM_001004198 | TATA box binding protein | Rn01455646_m1 |
| *Ubc* | NM_017314 | Ubiquitin C | Rn01789812_g1 |

### Analytical methods

RT-qPCR Ct values were acquired using the Applied Biosystems 7900HT Fast Real-Time PCR System software and relative gene expression calculated by the $2^{(-\Delta \mathrm{Ct})}$ method (Applied Biosystems Research Bulletin No. 2 P/N 4303859). Expression stability of putative reference genes was estimated using the BestKeeper (Technical University of Munich, Germany) and NormFinder (Aarhus University Hospital, Denmark) softwares.

BestKeeper make use of unconverted Ct values to perform parametric tests on normally distributed expression levels. It estimates the geometric mean of Ct values, determines the coefficient of variance (CV) for each gene, and calculates a Pearson's coefficient of correlation ($r$). Standard deviation (SD) calculations are performed to create a weighted index of most suitable normalizing genes across selected biological samples and exclude genes that are not stably expressed (*Pfaffl et al., 2004*). NormFinder uses Ct values transformed to a linear scale to estimate expression stability (S), combining intra- and inter-group variations for each reference gene. The algorithm takes into account biological heterogeneity and avoids selection bias caused by co-regulation of genes (*Andersen, Jensen & Ørntoft, 2004*).

## RESULTS

Before testing the expression stability of the 12 endogenous reference genes, all cDNA samples were normalised at the RNA level. Following RT-qPCR cycling, the raw mean Ct values of duplicate reactions were acquired, representing raw expression data. Ct values were used for statistical analyses.

From juvenile control rats, four biological replicates each of liver, fat pad, adrenal, prostate, testis and ovary tissues were analysed and the mean Ct values determined (Table 2). There was low inter-assay variation within the tissue types, with most standard deviation (SD) values being $<0.5$ and the highest 0.76. A larger variation was evident

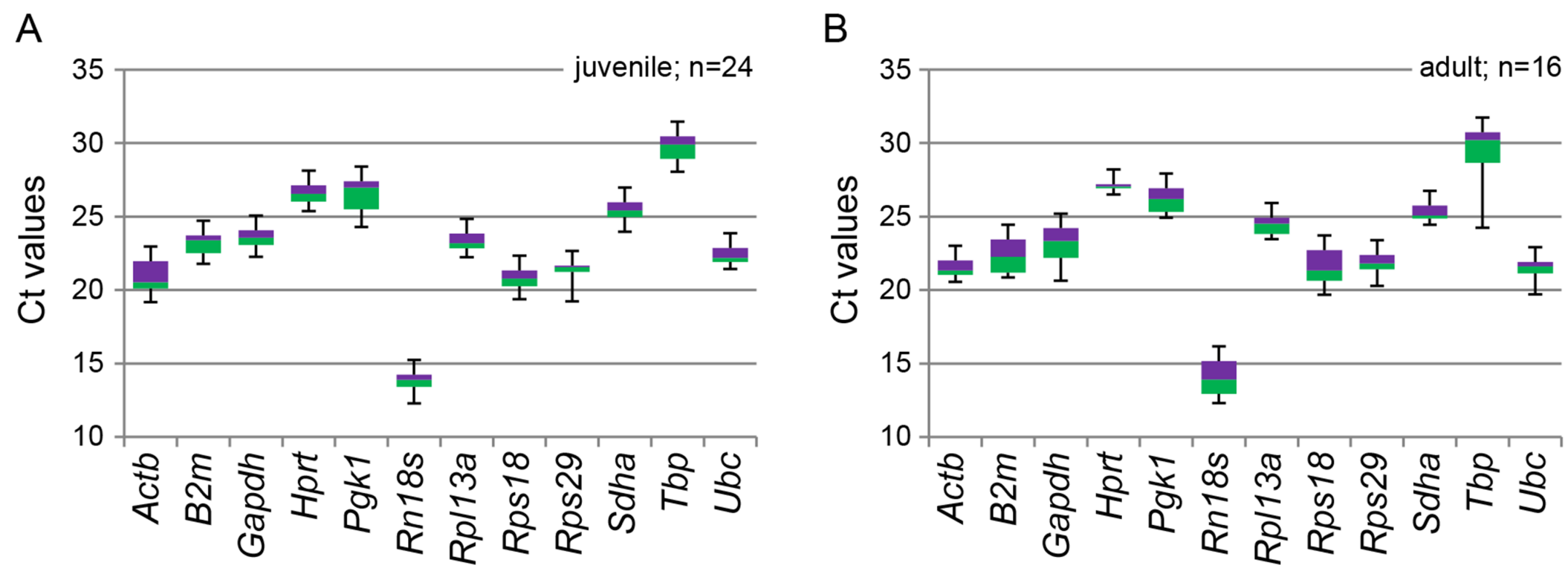

**Figure 1 Box plot representation of RT-qPCR threshold (Ct) values of 12 reference genes in rat tissues.** The reference genes were assayed across rat tissue cDNA samples, with each box plot representing the biological replicates. (A) Box plot of 24 juvenile rat tissues; 4 each of liver, fat pad, adrenal, prostate, testis and ovary. (B) Box plot of 16 adult rat tissues; 4 each of liver, fat pad, prostate and testis. Boxes denote median values with upper and lower quartiles, and whiskers minimum and maximum outliers.

**Table 2 Mean RT-qPCR threshold (Ct) values of 12 references genes in juvenile (13-17 days post natal) rat tissues.** Mean ± standard deviation (SD) was calculated from 4 biological replicates ($n = 4$).

| Gene | Liver (Mean ± SD) | Fat pad (Mean ± SD) | Adrenal (Mean ± SD) | Prostate (Mean ± SD) | Testis (Mean ± SD) | Ovary (Mean ± SD) | All (Mean ± SD) |
|---|---|---|---|---|---|---|---|
| ***Actb*** | 23.0 ± 0.23 | 22.0 ± 0.31 | 20.5 ± 0.31 | 20.0 ± 0.30 | 19.7 ± 0.69 | 20.4 ± 0.32 | 20.9 ± 1.25 |
| ***B2m*** | 23.6 ± 0.21 | 22.7 ± 0.19 | 22.1 ± 0.32 | 23.9 ± 0.26 | 23.9 ± 0.60 | 22.6 ± 0.51 | 23.2 ± 0.80 |
| ***Gapdh*** | 24.1 ± 0.19 | 24.2 ± 0.74 | 22.7 ± 0.52 | 23.9 ± 0.17 | 23.2 ± 0.32 | 23.2 ± 0.18 | 23.6 ± 0.69 |
| ***Hprt*** | 28.0 ± 0.37 | 27.1 ± 0.16 | 26.2 ± 0.35 | 26.7 ± 0.24 | 25.7 ± 0.26 | 26.0 ± 0.26 | 26.6 ± 0.81 |
| ***Pgk1*** | 27.9 ± 0.32 | 26.4 ± 0.57 | 24.7 ± 0.35 | 27.3 ± 0.39 | 27.3 ± 0.09 | 25.5 ± 0.21 | 26.5 ± 1.18 |
| ***Rn18s*** | 13.8 ± 0.45 | 12.5 ± 0.19 | 13.9 ± 0.33 | 14.2 ± 0.27 | 13.9 ± 0.65 | 14.3 ± 0.38 | 13.8 ± 0.69 |
| ***Rpl13a*** | 25.4 ± 0.17 | 23.8 ± 0.12 | 23.2 ± 0.18 | 23.1 ± 0.26 | 22.9 ± 0.06 | 22.3 ± 0.06 | 23.5 ± 1.02 |
| ***Rps18*** | 22.3 ± 0.33 | 21.4 ± 0.18 | 20.8 ± 0.11 | 20.5 ± 0.33 | 20.5 ± 0.24 | 19.6 ± 0.22 | 20.9 ± 0.88 |
| ***Rps29*** | 21.6 ± 0.24 | 21.5 ± 0.12 | 21.9 ± 0.12 | 21.3 ± 0.18 | 21.2 ± 0.76 | 19.3 ± 0.11 | 21.1 ± 0.91 |
| ***Sdha*** | 26.5 ± 0.26 | 25.6 ± 0.19 | 24.3 ± 0.36 | 26.0 ± 0.21 | 25.1 ± 0.11 | 25.1 ± 0.38 | 25.4 ± 0.76 |
| ***Tbp*** | 32.7 ± 0.31 | 29.8 ± 0.23 | 30.2 ± 0.37 | 30.1 ± 0.31 | 28.2 ± 0.23 | 28.9 ± 0.13 | 30.0 ± 1.43 |
| ***Ubc*** | 23.9 ± 0.26 | 23.0 ± 0.26 | 22.3 ± 0.29 | 21.7 ± 0.16 | 22.0 ± 0.18 | 22.0 ± 0.14 | 22.5 ± 0.78 |

between tissue types, with SD values across all the 24 tissue samples being >0.5 and that of *Actb*, *Pgk1*, *Rpl13a*, and *Tbp* exceeding 1.0. The expression range of the individual reference genes are represented as box plots in Fig. 1A.

From adult control rats, four biological replicates each of liver, fat pad, prostate and testis tissues were analysed and the mean Ct values determined (Table 3). Again, there was low inter-assay variation within tissue types for most of the reference genes, with most SD values being <0.5. One exception was *Rn18s* where SD values were >1.5 in liver and testis, and 0.94 in prostate. Between different tissues, the variation in Ct values was larger. Across all the 16 tissue samples (4 tissue types) the SD was >0.5 for all reference genes, with *B2m*, *Gapdh*, *Rn18s*, *Rps18* and *Tbp* exceeding 1.0. Notably, *Tbp* displayed a significantly lower Ct

**Table 3 Mean RT-qPCR threshold (Ct) values of 12 reference genes in adult rat tissues.** Mean ± standard deviation (SD) was calculated from 4 biological replicates ($n = 4$).

| Gene | Liver (Mean ± SD) | Fat pad (Mean ± SD) | Prostate (Mean ± SD) | Testis (Mean ± SD) | All (Mean ± SD) |
|---|---|---|---|---|---|
| *Actb* | 23.1 ± 0.16 | 21.3 ± 0.16 | 21.1 ± 0.49 | 21.0 ± 0.30 | 21.6 ± 0.92 |
| *B2m* | 21.4 ± 0.38 | 21.1 ± 0.27 | 23.9 ± 0.81 | 23.3 ± 0.26 | 22.4 ± 1.33 |
| *Gapdh* | 20.7 ± 0.12 | 23.0 ± 0.25 | 24.6 ± 0.25 | 23.7 ± 0.31 | 23.0 ± 1.48 |
| *Hprt* | 27.7 ± 0.39 | 27.1 ± 0.11 | 26.8 ± 0.26 | 27.0 ± 0.17 | 27.1 ± 0.43 |
| *Pgk1* | 25.8 ± 0.35 | 25.2 ± 0.18 | 26.9 ± 0.47 | 26.9 ± 0.32 | 26.2 ± 0.81 |
| *Rn18s* | 14.8 ± 1.55 | 12.7 ± 0.28 | 14.7 ± 0.94 | 14.4 ± 1.56 | 14.1 ± 1.38 |
| *Rpl13a* | 25.6 ± 0.35 | 24.3 ± 0.40 | 23.6 ± 0.16 | 24.5 ± 0.23 | 24.5 ± 0.79 |
| *Rps18* | 23.1 ± 0.82 | 22.3 ± 0.75 | 21.0 ± 0.27 | 19.8 ± 0.21 | 21.5 ± 1.38 |
| *Rps29* | 23.2 ± 0.25 | 22.0 ± 0.38 | 21.1 ± 0.68 | 21.5 ± 0.29 | 22.0 ± 0.90 |
| *Sdha* | 25.0 ± 0.42 | 24.7 ± 0.19 | 26.4 ± 0.22 | 25.2 ± 0.20 | 25.3 ± 0.69 |
| *Tbp* | 31.6 ± 0.61 | 30.2 ± 0.25 | 30.3 ± 0.25 | 24.8 ± 0.40 | 29.2 ± 2.73 |
| *Ubc* | 22.1 ± 0.42 | 21.8 ± 0.17 | 21.5 ± 0.16 | 20.1 ± 0.30 | 21.4 ± 0.84 |

value in adult testis (Ct = 24.8) than in any other adult or juvenile tissues (Ct = 28.2-32.7). The expression range of the individual reference genes across adult tissues are given as box plots in Fig. 1B.

Reference genes would be expected to display most stable transcript abundance between samples of the same tissues and organs and within similar developmental stages. The greatest variations to expression levels would be expected between different tissue types, at different developmental stages, but also if they have been affected by external factors. Here, we included RT-qPCR analyses of tissues from rats having been exposed to mixtures of chemical compounds. These tissue samples correlated with the control tissues above. The first group comprised juvenile ovaries and prostate, as well as adult prostate tissues obtained from rats having been exposed peri- and postnatally to Mix450 (Table 4). Some smaller variations in Ct values were observed between tissues from those of the control animals (Tables 2 and 3), but did not exceed 0.5 cycles. The second group comprised adult liver, fat pad and testis tissues obtained from male rats having been exposed postnatally to either PFNA or PFNA/mix (Table 5). Here, greater variability of Ct values compared to tissues from control animals (Table 3) was evident. In the liver, the Ct values from several genes varied by more than 1.0 cycle, including *Gapdh*, *Hprt*, *Pgk1*, *Rps29*, *Rpl13a* and *Rps18*, the latter two with as much as 2.6 and 3.1 cycles, respectively. In the fat pad the difference in expression was far less, with only *Gapdh* in PFNA-exposed rats exceeding 1.0 cycle difference relative to control. In the testis, most changes to Ct values in exposed animals versus control was less than 1.0 cycle, but with *Gapdh* varying by 1.5 cycles in PFNA/mix and *Rn18s* as much as 2.2 cycles in the PFNA group.

To assess relative expression stability across different biological samples, we analysed the Ct outputs with NormFinder (*Andersen, Jensen & Ørntoft, 2004*). First, raw Ct values were converted to linear scale by the $2^{(-\Delta Ct)}$ method using the geometric mean of *Hprt*, *Pgk1*, *Rpl13a*, *Rps18*, *Rps29*, *Sdha* and *Ubc* based on initial determination of coefficient

**Table 4 Mean RT-qPCR threshold (Ct) values of 12 reference genes in juvenile (J) and adult (A) tissues from rats exposed to Mix450.** Mean ± standard deviation (SD) was calculated from 4 biological replicates ($n = 4$).

| Gene | Ovary (J) + Mix450 (Mean ± SD) | Prostate (J) + Mix450 (Mean ± SD) | Prostate (A) + Mix450 (Mean ± SD) |
|---|---|---|---|
| *Actb* | 20.6 ± 0.45 | 20.1 ± 0.19 | 21.4 ± 0.12 |
| *B2m* | 23.0 ± 0.47 | 23.8 ± 0.15 | 24.4 ± 0.29 |
| *Gapdh* | 23.2 ± 0.10 | 24.1 ± 0.36 | 24.2 ± 0.19 |
| *Hprt* | 26.2 ± 0.15 | 26.6 ± 0.15 | 26.9 ± 0.06 |
| *Pgk1* | 25.6 ± 0.28 | 27.1 ± 0.15 | 26.5 ± 0.07 |
| *Rn18s* | 14.1 ± 0.16 | 14.5 ± 0.56 | 14.8 ± 0.12 |
| *Rpl13a* | 22.7 ± 0.11 | 22.8 ± 0.26 | 23.7 ± 0.04 |
| *Rps18* | 19.8 ± 0.19 | 20.1 ± 0.31 | 21.0 ± 0.09 |
| *Rps29* | 19.5 ± 0.04 | 21.2 ± 0.25 | 21.6 ± 0.05 |
| *Sdha* | 25.1 ± 0.28 | 25.8 ± 0.25 | 26.3 ± 0.16 |
| *Tbp* | 29.2 ± 0.35 | 29.8 ± 0.19 | 30.4 ± 0.08 |
| *Ubc* | 22.4 ± 0.22 | 21.6 ± 0.29 | 21.3 ± 0.06 |

**Table 5 Mean RT-qPCR threshold (Ct) values of 12 reference genes in adult tissues from rats exposed to hPFNA or hPFNA/Mix.** Mean ± standard deviation (SD) was calculated from 4 biological replicates ($n = 4$.)

| Gene | Liver + hPFNA (Mean ± SD) | Liver + hPFNA/mix (Mean ± SD) | Fat pad + hPFNA (Mean ± SD) | Fat pad + hPFNA/mix (Mean ± SD) | Testis + hPFNA (Mean ± SD) | Testis + hPFNA/mix (Mean ± SD) |
|---|---|---|---|---|---|---|
| *Actb* | 22.8 ± 0.06 | 22.5 ± 0.37 | 21.6 ± 0.30 | 21.6 ± 0.34 | 20.3 ± 0.25 | 20.7 ± 0.29 |
| *B2m* | 21.4 ± 0.12 | 21.5 ± 0.27 | 20.8 ± 0.15 | 21.2 ± 0.53 | 23.6 ± 0.47 | 23.8 ± 0.44 |
| *Gapdh* | 19.8 ± 0.18 | 19.7 ± 0.16 | 24.1 ± 0.53 | 23.5 ± 0.72 | 23.9 ± 0.30 | 22.2 ± 1.34 |
| *Hprt* | 26.5 ± 0.27 | 26.4 ± 0.17 | 26.9 ± 0.18 | 26.7 ± 0.81 | 26.9 ± 0.30 | 27.2 ± 0.24 |
| *Pgk1* | 24.3 ± 0.24 | 24.1 ± 0.26 | 25.6 ± 0.34 | 25.4 ± 0.33 | 27.3 ± 0.45 | 27.8 ± 0.42 |
| *Rn18s* | 15.1 ± 1.27 | 14.1 ± 0.17 | 12.6 ± 0.27 | 13.4 ± 1.10 | 12.2 ± 0.04 | 14.1 ± 0.89 |
| *Rpl13a* | 23.1 ± 0.19 | 23.0 ± 0.33 | 23.8 ± 0.29 | 23.8 ± 0.31 | 24.4 ± 0.33 | 24.5 ± 0.21 |
| *Rps18* | 20.2 ± 0.21 | 20.0 ± 0.37 | 21.4 ± 0.33 | 21.5 ± 0.25 | 19.7 ± 0.26 | 19.9 ± 0.25 |
| *Rps29* | 21.6 ± 0.18 | 21.5 ± 0.28 | 21.8 ± 0.32 | 21.8 ± 0.23 | 21.3 ± 0.28 | 21.5 ± 0.22 |
| *Sdha* | 24.3 ± 0.28 | 24.1 ± 0.21 | 24.8 ± 0.33 | 24.6 ± 0.53 | 24.9 ± 0.25 | 25.0 ± 0.31 |
| *Tbp* | 30.5 ± 0.41 | 30.2 ± 0.23 | 29.8 ± 0.14 | 29.9 ± 0.09 | 24.3 ± 0.33 | 24.5 ± 0.33 |
| *Ubc* | 21.0 ± 0.21 | 20.8 ± 0.24 | 21.8 ± 0.08 | 21.4 ± 0.54 | 19.8 ± 0.39 | 20.0 ± 0.51 |

of variance (CV) across the biological replicates (data not shown). Based on converted Ct values, the software calculates a stability value (S) for each gene across the selected biological replicates. First we assessed expression stability across control tissues, which included the 40 biological replicates listed in Tables 2 and 3 (Fig. 2A). Here, *Hprt* and *Sdha* were ranked as the most stable, whereas *Tbp* and *B2m* were deemed least stable. NormFinder recommends an upper S-value of 0.5 for genes to be assessed as relatively stable. Only *Hprt* and *Sdha* had an S-value <0.5, but *Ubc*, *Actb*, *Rpl13a* and *Rps18* were all close to 0.5. A further 36 samples were included in the NormFinder assessment, which

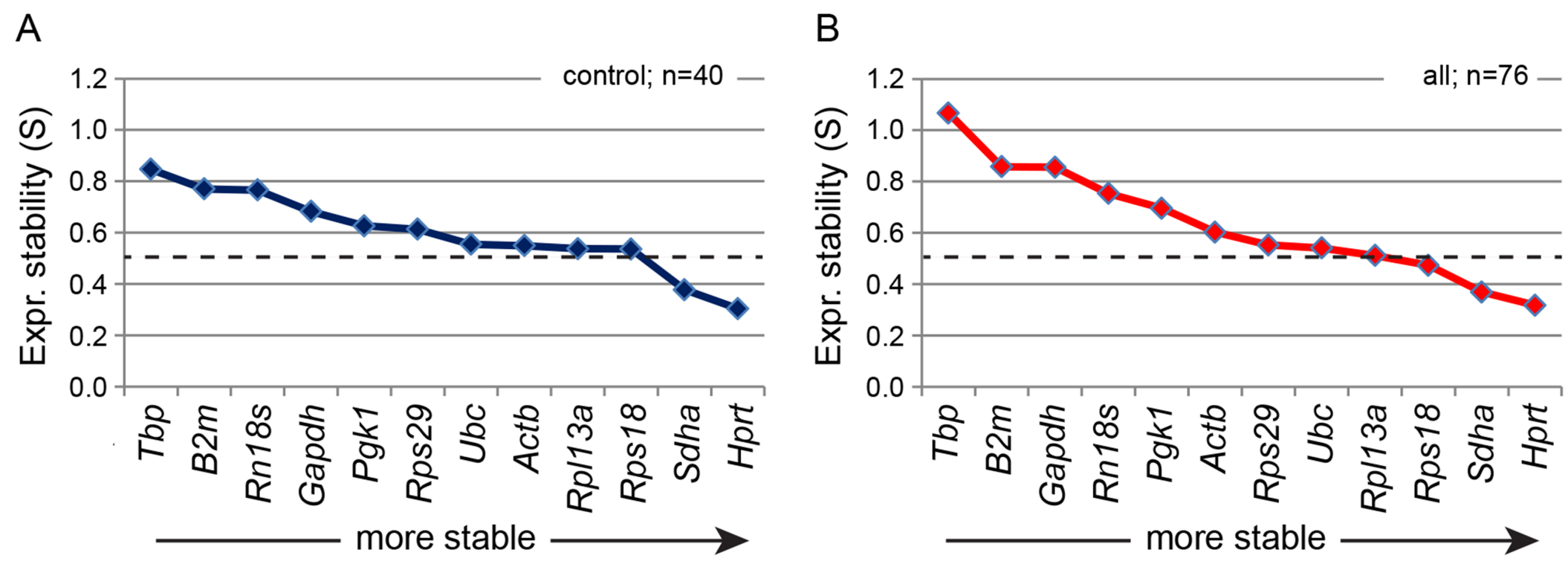

**Figure 2 Relative expression stability (S) of 12 reference genes across rat tissues as determined by NormFinder.** The 12 reference genes were assayed across (A) control tissues ($n = 40$) from juvenile and adult rats and (B) the control tissues plus adult tissues from rats subjected to chemical exposures ($n = 76$). The tissues included prostate, testis, adrenal, liver, fat, and ovary. Higher S-values represent lower expression stability, with $S < 0.5$ typically indicating relative stability suitable for use as endogenous normalization. The data show a shift towards higher S-values (B vs. A) for the most unstable gene transcripts following chemical exposure.

**Table 6 Expression stability between juvenile tissue types according to relative NormFinder stability values.** Shaded boxes indicate a stability value $(S) > 0.5$, thus deemed unstable within the specific comparisons.

| | Liver *vs* fat pad | Liver *vs* testis | Adrenal *vs* prostate | Prostate *vs* fat pad | Adrenal *vs* ovary | Ovary *vs* testis |
|---|---|---|---|---|---|---|
| **Most stable** | *B2m* | *Sdha* | *Hprt* | *Rps29* | *Hprt* | *Rpl13a* |
| | *Sdha* | *Rps18* | *Rn18s* | *Hprt* | *Ubc* | *Gapdh* |
| | *Actb* | *Ubc* | *Rpl13a* | *Tbp* | *Actb* | *Ubc* |
| | *Rps18* | *Hprt* | *Tbp* | *Sdha* | *Rpl13a* | *Sdha* |
| | *Hprt* | *Gapdh* | *Rps18* | *Gapdh* | *Rn18s* | *Rps18* |
| | *Rn18s* | *Rpl13a* | *Gapdh* | *Rpl13a* | *Rps18* | *Hprt* |
| | *Ubc* | *Pgk1* | *Actb* | *Rps18* | *Gapdh* | *Tbp* |
| | *Rpl13a* | *Rps29* | *Rps29* | *Pgk1* | *B2m* | *Rn18s* |
| | *Pgk1* | *B2m* | *Sdha* | *Ubc* | *Tbp* | *B2m* |
| | *Rps29* | *Actb* | *Ubc* | *B2m* | *Pgk1* | *Actb* |
| **Least stable** | *Gapdh* | *Rn18s* | *B2m* | *Rn18s* | *Sdha* | *Pgk1* |
| | *Tbp* | *Tbp* | *Pgk1* | *Actb* | *Rps29* | *Rps29* |

included tissues from adult rats that had been exposed to chemical mixtures during development (Fig. 2B). The most stably expressed genes *Hprt* and *Sdha* (as determined above) appeared relatively unchanged in these tissues, whereas more unstably expressed genes shifted to a significantly higher S-value. This was particularly evident for *Tbp*, *B2m*, *Gapdh* and *Rn18s*. NormFinder was also used to assess pairwise expression stabilities of the 12 reference genes between juvenile rat tissues. Although it represents only a selection of possible pairing of tissues, it illustrates that optimal reference genes can vary significantly between tissue types (Table 6).

Next, relative expression stability was assessed by the BestKeeper software (*Pfaffl et al., 2004*), which defined genes that display a SD >1.0 as unstable. Based on the NormFinder

**Table 7 BestKeeper correlation analysis of 10 reference genes in rat tissues.** Shaded boxes indicate genes deemed unsuitable as reference genes based on too high SD or *P* values.

| Control tissues; $n = 40$. | | | | | All tissues; $n = 76$. | | | | |
|---|---|---|---|---|---|---|---|---|---|
| Gene | SD | CV | *r* | *P* | Gene | SD | CV | *r* | *P* |
| *Rn18s* | 0.5438 | 3.9513 | −0.222 | 0.296 | *Hprt* | 0.4612 | 1.7226 | 0.747 | 0.001 |
| *Gapdh* | 0.5597 | 2.3766 | 0.634 | 0.001 | *Rps29* | 0.5647 | 2.6401 | 0.483 | 0.001 |
| *Sdha* | 0.6169 | 2.4245 | 0.778 | 0.001 | *Sdha* | 0.6027 | 2.3911 | 0.679 | 0.001 |
| *Ubc* | 0.6408 | 2.8510 | 0.797 | 0.001 | *Rpl13a* | 0.7404 | 3.1224 | 0.674 | 0.001 |
| *Hprt* | 0.6792 | 2.5506 | 0.927 | 0.001 | *Ubc* | 0.7584 | 3.5106 | 0.501 | 0.001 |
| *Rps18* | 0.6827 | 3.2732 | 0.911 | 0.001 | *Rps18* | 0.8150 | 3.9201 | 0.747 | 0.001 |
| *Rps29* | 0.7000 | 3.3100 | 0.467 | 0.021 | *Rn18s* | 0.8260 | 5.9426 | 0.230 | 0.046 |
| *Rpl13a* | 0.7822 | 3.3351 | 0.960 | 0.001 | *Actb* | 0.8794 | 4.1409 | 0.467 | 0.001 |
| *Pgk1* | 1.0420 | 3.9300 | 0.666 | 0.001 | *Pgk1* | 1.0071 | 3.8459 | 0.476 | 0.001 |
| *Actb* | 1.0472 | 5.0032 | 0.841 | 0.001 | *Gapdh* | 1.1385 | 4.9379 | 0.310 | 0.007 |

**Notes.**
Control, tissues from unexposed animals; All, tissues from unexposed and exposed animals.

output, we excluded *Tbp* and *B2m* from these analyses, only assessing the remaining 10 reference genes. The analyses were first performed with the 40 control samples listed in Tables 2 and 3. Eight of the genes displayed a SD $<1.0$, with the remaining two, *Pgk1* and *Actb*, SD $>1.0$ (Table 7). *Rn18s* displayed the lowest SD, but with $P = 0.296$ it is not possible to conclude on stability. When including an additional 36 tissue samples from rats exposed to chemical mixtures (Tables 4 and 5), the stability ranking of the genes altered. Again, eight genes had an SD $<1.0$, but *Pgk1* and *Gapdh* had an SD $>1.0$, thus excluding them as suitable reference genes for these tissue samples (Table 7).

## DISCUSSION

The use of a suitable reference when normalizing RT-qPCR data is essential to obtain data that truthfully represents relative transcript abundance of genes within cells and tissues. The most common strategy is to use endogenous reference genes, but unfortunately the chosen reference genes have often not been properly verified as being stably expressed across the samples being analysed. Here, we have analysed a panel of 12 commonly used reference genes across various tissues from juvenile and adult rats and recommend what reference genes to use for these tissues based on empirical data. Further guidance on how to monitor and select suitable reference genes for future rat studies with variable experimental parameters is given.

When looking at specific tissues, the relative expression levels of reference genes are normally quite stable between biological samples. Significant variability in RNA transcript abundance is often first encountered when comparing the same tissue type from different developmental stages, when comparing different types of tissues, or when severe adverse effects are induced (*Cabiati et al., 2012*; *DuBois et al., 2013*; *Hvid et al., 2011*; *Kim et al., 2011*; *Martínez-Beamonte et al., 2011*; *Svingen et al., 2009*). Here, we found several reference genes to be relatively stable across juvenile and adult rat tissues; however, none of the 12

genes showed variations Ct $<1.0$ when assessed across all 10 different biological groups. For example, *Hprt*, which generally showed the greatest stability across tissues, displayed lower expression levels (higher Ct) in liver than in any other tissue. Also *Actb*, *Rpl13a*, *Rps18* and *Tbp* showed a similar trend of being expressed at a lower level in liver than in other tissues. *Rn18s* was generally stable across tissues with the exception of the fat pad where relative expression typically was higher (lower Ct by approx. 2.0) than for other tissues, which could suggests that the rRNA/mRNA ration is skewed between these tissues. Another example is the higher expression of *Tbp* in juvenile testis and ovary, compared to other tissues and an even higher expression in adult testis (Ct $>5.0$ difference). *Tbp* has previously been shown to be a suitable reference gene in fetal mouse gonads, as well as human fetal testis (*O'Shaughnessy, Monteiro & Fowler, 2011*; *Svingen et al., 2009*), which highlights the importance of not simply relying on data from one species (e.g., mouse) when selecting suitable reference genes for another (e.g., rat).

An additional problem is encountered when normalizing gene expression in experimentally manipulated tissues; in the context of this study, animals that had been exposed to a variety of chemicals, often at toxicological dose levels. Experimental manipulations can adversely affect gene expression levels in various tissues, including genes normally regarded as 'stable housekeeping genes' (*Das, Banerjee & Shapiro, 2013*; *DuBois et al., 2013*; *Martínez-Beamonte et al., 2011*; *Nair et al., 2014*; *Pohjanvirta et al., 2006*; *Santin et al., 2013*; *Silver et al., 2008*; *Taki, Abdel-Rahman & Zhang, 2014*). Here, we have also shown that the expression of endogenous reference genes such as *Tbp*, *B2m* and *Gapdh* are affected in tissues having been exposed to chemicals during development. Although most changes in relative transcript abundance were less than one cycle (Ct $<1.0$), some tissues displayed higher variability following chemical exposure. Not surprisingly the liver was most affected, with for instance *Pgk1*, *Rpl13a* and *Rps18* shifting as much as Ct $>1.5$, $>2.5$ and $>3.0$, respectively in animals having been exposed to high PFNA levels with or without a background mix of chemicals (Table 3 versus Table 5). In numerical terms, this could translate into a nearly 10-fold difference in reported change in relative fold gene expression. As reported previously (*Hadrup et al., 2015*), the PFNA exposure was modelled on high-end human relevant exposure and animals were affected to variable degree. For instance, the livers from exposed rats displayed some hypertrophy and steatosis. Other organs did not show significant effects at a macroscopic or histological level, but high doses resulted in decreased levels of systemic androgens (testosterone) and increased aromatase expression in fatty tissues. There was also an observable downregulation of steroidogenic genes in the testes from high-dosed rats, including *StAR*, *Cyp11a1*, *Cyp17a1* and *Hsd17b1*, suggestive of a decreased steroidogenic output.

Previous studies have reported on the potential to generate flawed data by selecting inappropriate reference genes (*Dheda et al., 2005*; *Svingen, Jørgensen & Rajpert-De Meyts, 2014*). Therefore, the suitability of employed reference genes must be empirically verified for all individual experiments. This is further exemplified by these tissue samples, where animals having been exposed to chemicals show altered expression of constitutively expressed 'housekeeping' genes, both in tissues with obvious pathology, but also tissues

where no histopathological effects are discernible. But an additional aspect to consider in toxicological studies is also the potential induction of widespread necrosis that could compromise the overall quality of the RNA pool.

In terms of our data set, several of the analysed reference genes can be successfully employed for RT-qPCR analysis. NormFinder suggests that both *Hprt* and *Sdha* are relatively stable across all the tissues examined, including after chemical exposure. Other genes were scored as borderline suitable and included *Rps18*, *Rpl13a* and *Ubc*, but with *Tbp*, *B2m*, *Gapdh* and *Rn18s* not suitable. Contradictorily the BestKeeper software suggested all genes except *Pgk1* and *Actb* to be stable enough for normalization in the control tissues. A discrepancy in stability rankings and outcomes between different analytical methods such as NormFinder and BestKeeper has been reported before and are attributed to the different theoretical algorithms that are employed (*Andersen, Jensen & Ørntoft, 2004*; *Martínez-Beamonte et al., 2011*; *Pfaffl et al., 2004*). Nevertheless, they are often in overall agreement, which was also the case with our data with the exception of *Gapdh* and *Rn18s* in control tissues. For *Rn18s*, a $p$-value of 0.3 excludes the possibility of determining with certainty if it is stably expressed. But more importantly, the NormFinder algorithm shows a bias towards a variant gene if several other genes display a systematic variation across samples, which is likely the case for *Rn18s* in these tissues. Finally, a gene does not have to rank on top of the list to be suitable for normalization purposes as long as they perform better than the exclusion criteria. Further, it is recommended to use the geometric mean of two or more reference genes rather than relying on a single gene.

Although screening all tissues under all experimental conditions when selecting suitable reference genes is recommended, it is not always practically feasible. Therefore, we suggest a monitoring protocol after initial screening of selected tissues, then re-screen when appropriate. Firstly, we reiterate the importance of always normalising tissue samples at the RNA level; that is, using a stable input amount of RNA for all cDNA synthesis reaction, as variation in reverse transcription protocols can significantly affect downstream RT-qPCR assays (*Ståhlberg et al., 2004*; *Bustin et al., 2015*). Secondly, after selecting 2 or more reference genes that are deemed stable in the tissues to be examined, monitor for any significant changes in Ct values. If Ct values change by more than 2 cycles between biological replicates, selection of other reference genes should be considered; after additional screening tests if necessary. Thirdly, to adjust for smaller changes in expression of the reference genes, the use of a geometric mean from at least three reference genes to normalize data has been recommended (*Vandesompele et al., 2002*). In this manner, significant changes to the expression of reference genes following experimental conditions can be detected, ultimately avoiding reporting of significantly flawed data.

## ACKNOWLEDGEMENTS

We like to thank the staff at the animal facilities at the National Food Institute (Technical University of Denmark) for husbandry and help with tissue collection.

## Funding

This research was supported by a grant from the European Commission, 7th Framework Programme, CONTAMED (no.: 212502) and the Ministry of Food, Agriculture and Fisheries. The funders had no role in study design, data collection and analysis, decision to publish, or preparation of the manuscript.

## Grant Disclosures

The following grant information was disclosed by the authors:
European Commission 7th Framework Programme, CONTAMED: 212502.
Ministry of Food, Agriculture and Fisheries.

## Competing Interests

The authors declare there are no competing interests.

## Author Contributions

- Terje Svingen conceived and designed the experiments, performed the experiments, analyzed the data, wrote the paper, prepared figures and/or tables, reviewed drafts of the paper.
- Heidi Letting conceived and designed the experiments, performed the experiments, analyzed the data, reviewed drafts of the paper.
- Niels Hadrup conceived and designed the experiments, reviewed drafts of the paper.
- Ulla Hass and Anne Marie Vinggaard conceived and designed the experiments, contributed reagents/materials/analysis tools, reviewed drafts of the paper.

## Animal Ethics

The following information was supplied relating to ethical approvals (i.e., approving body and any reference numbers):

Experimental protocols and use of animals were approved by the Danish Animal Experiments Inspectorate (Permit No. 2012-15-2934-00089 C4) and overseen by the Animal Welfare Committee of the National Food Institute, Technical University of Denmark.

## Supplemental Information

Supplemental information for this article can be found online at http://dx.doi.org/10.7717/peerj.855#supplemental-information.

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
