# Peer review of "Selection of reference genes for quantitative RT-PCR (RT-qPCR) analysis of rat tissues under physiological and toxicological conditions"

_PeerJ, doi:10.7717/peerj.855_

## Round 0.1 · original submission · Minor Revisions

Thanks for sharing this nice work. It seems it needs only some minor revisions. Please be careful in your use of the term "evidence-based" as it is typically understood to refer to evidence-based medicine and I do not have the impression that this is what you mean.

·

Basic reporting

In Table 4 the expression ‘normalising genes’ is used instead of ‘reference genes’ as else in the text, be consistent.

Experimental design

The design of the Rn18s could influence the results of this gene when compared to the TaqMan Gene Assays purchased from Life Technologies. Why not make use of the Rn18s from Life Technologies as well?

Validity of the findings

Some information on the observed toxic effects (from Hadrup et al 2015), preferable gene expression data, in the exposed animals would enhance the conclusions of the paper. Variation of reference genes in correlation with changes observed in genes of interest would give a better understanding of the stability of the reference genes. At least a sentence about the toxicity would be important to prove that there is a difference between non-exposed and exposed animals.

Additional comments

Housekeeping genes and normalization of genes are important issues, addressed by the authors. I welcome such a paper and hope that the community will improve the reporting and validation of selected reference genes in future peer-reviewed articles.

·

Basic reporting

No Comments

Experimental design

No comments

Validity of the findings

No comments

Additional comments

While of experimental settings develop along with increasing demand for testing new drugs and toxic compounds, evaluating and standardization of methods for finding and quantifying biomarkers are especially important. One of the widely used methods which requires standardization is RT-qPCR since the growing body of evidence have indicated deficiency of existing strategies normalizing RT-qPCR data. The authors address this issue in elegant way by
analyseing the expression stability of 12 commonly used reference genes (Actb, B2m, Gapdh, Hprt, Pgk1, Rn18s, Rpl13a, Rps18, Rps29, Sdha, Tbp and Ubc) across several juvenile and adult rat tissues. Some reference genes were shown to be unstable and this effect was tissue specific. To standardized utilization of RT-qPCR method the authors have proposed an additional normalisation at the RNA level, and continuous monitoring of the suitability of selected reference genes under various experimental conditions.
This article, though mainly methodological is very important both for gathering and interpretation of the RT-qPCR. The manuscript contain: 1) comprehensive introduction providing the actual state of the art, 2) well and correctly described methods, 3) clear an convincing results, 4) the mature discussion. Finally it provides clear roadmap how to select reference genes and plane reliable experimentation using RT-qPCR method. I highly recommend publication of this paper.

Reviewer 3 ·

Basic reporting

The manuscript of Svingen et al. is well written and addresses an overlooked aspect in the quantification of gene expression levels in rodents, particularly in the field of endocrine-targeted tissues (mainly reproductive). Although some relevant observations are presented and discussed, some conclusions deserve to be discussed in view of both: 1. previously well established assessment on the use of multiple housekeeping genes to set up a proper normalization of gene expression data threshold; and 2. usual (hopefully) good laboratory practises.

Experimental design

Major concern 1. All over the manuscript it is stated that the relative gene expression has been calculated by the 2(-deltaCt) method but in Methods and/or Result sections is lacking any discussed consideration about the calculation of the qPCR efficiency of each tested gene, although the authors mentioned the used softwares and the publication (Pfaffl et al. 2004) in which the application range of the above method is detailed and discussed.
Major concern 2. Within the Abstract ("additional normalisation at the RNA level") and Discussion (lines 268-270), the authors recommend "a stable input amount of RNA for all cDNA synthesis reactions". Honestly speaking, both sentences are somehow astonishing and it suggests that usually it is not done either in your laboratory or in any other laboratories. That it might also be although I would never accept it. Hence, I would recommend to change such statements and showing (and discuss) also in your results that you observed differences in gene expression levels if you change your stable RNA input that in my experience is the first assessment that must be done along with the calculation of the qPCR efficiency. Do the authors normally do that? Could you provide a prove of it?

Validity of the findings

Major concern 3. Within Introduction (lines 45-46), the authors stated the importance about transcript abundance that should not be considered the same in all cells. But no where else is mentioned the fact that the selected, proper housekeeping genes should have a similar transcript abundance to the one(s) of interest in order to avoid under- or over-estimation of relative abundance or on the way to bypass such potential misleading effects. Do the authors considered to take into account and discuss these aspects (see Vandesompele et al. Genome Biology 2002 and related publications) before to "partially" conclude that at least two or more housekeeping genes are the best choice (Discussion lines 263-264 and 271-275) just on the view that they are stably abundant in different cells/tissues? Using similar starting point considerations in the Vandesompele et al. manuscript, it was concluded that at least three housekeeping genes have to be considered for a proper normalization. And their approach was evidence-based as the authors claimed in Introduction lines 75-77.

---

## Round 0.2 · accepted · Accept

Thank you very much for addressing the suggestions of the reviewers in an appropriate way.